# Is *CYP2C* Haplotype Relevant for Efficacy and Bleeding Risk in Clopidogrel-Treated Patients?

**DOI:** 10.3390/genes15050607

**Published:** 2024-05-10

**Authors:** Lana Ganoci, Jozefina Palić, Vladimir Trkulja, Katarina Starčević, Livija Šimičević, Nada Božina, Martina Lovrić-Benčić, Zdravka Poljaković, Tamara Božina

**Affiliations:** 1Division of Pharmacogenomics and Therapy Individualization, Department of Laboratory Diagnostics, University Hospital Centre Zagreb, 10000 Zagreb, Croatia; lana.ganoci@kbc-zagreb.hr (L.G.);; 2Department of Pharmacology, School of Medicine, University of Zagreb, 10000 Zagreb, Croatia; vladimir.trkulja@mef.hr (V.T.); nada.bozina@mef.hr (N.B.); 3Department of Medical Chemistry, Biochemistry and Clinical Chemistry, School of Medicine, University of Zagreb, 10000 Zagreb, Croatia; jozefina.palic@mef.hr; 4Department of Neurology, University Hospital Centre Zagreb, 10000 Zagreb, Croatia; katarina.starcevic@kbc-zagreb.hr (K.S.); zdravka.poljakovic@kbc-zagreb.hr (Z.P.); 5School of Medicine, University of Zagreb, 10000 Zagreb, Croatia; martina.lovric-bencic@kbc-zagreb.hr; 6Department of Cardiovascular Diseases, University Hospital Centre Zagreb, 10000 Zagreb, Croatia

**Keywords:** clopidogrel, CYP2C19 phenotype, *CYP2C:TG* haplotype, efficacy, bleeding

## Abstract

A recently discovered haplotype—*CYP2C:TG*—determines the ultrarapid metabolism of several CYP2C19 substrates. The platelet inhibitor clopidogrel requires CYP2C19-mediated activation: the risk of ischemic events is increased in patients with a poor (PM) or intermediate (IM) CYP2C19 metabolizer phenotype (vs. normal, NM; rapid, RM; or ultrarapid, UM). We investigated whether the *CYP2C:TG* haplotype affected efficacy/bleeding risk in clopidogrel-treated patients. Adults (*n* = 283) treated with clopidogrel over 3–6 months were classified by CYP2C19 phenotype based on the *CYP2C19*2*17* genotype, and based on the *CYP2C19/CYP2C* cluster genotype, and regarding carriage of the *CYP2:TG* haplotype, and were balanced on a number of covariates across the levels of phenotypes/haplotype carriage. Overall, 45 (15.9%) patients experienced ischemic events, and 49 (17.3%) experienced bleedings. By either classification, the incidence of ischemic events was similarly numerically higher in PM/IM patients (21.6%, 21.8%, respectively) than in mutually similar NM, RM, and UM patients (13.2–14.8%), whereas the incidence of bleeding events was numerically lower (13.1% vs. 16.6–20.5%). The incidence of ischemic events was similar in *CYP2C:TG* carries and non-carries (14.1% vs. 16.1%), whereas the incidence of bleedings appeared mildly lower in the former (14.9% vs. 20.1%). We observed no signal to suggest a major effect of the *CYP2C19/CYP2C* cluster genotype or *CYP2C:TG* haplotype on the clinical efficacy/safety of clopidogrel.

## 1. Introduction

The cytochrome P-450 isoenzyme CYP2C19 (CYP2C19) is encoded by a highly polymorphic *CYP2C19* gene: more than 39 star (*) allele haplotypes have been defined by the Pharmacogene Variation Consortium [1,2]. Alleles greatly reflect on the enzyme activity, and are categorized into functional groups: normal function alleles (e.g., *CYP2C19*1*), decreased function alleles (e.g., *CYP2C19*9*), no function alleles (e.g., *CYP2C19*2* and **3*), and increased function alleles (e.g., *CYP2C19*17*) [2,3]. Based on the genotype, five CYP2C19 metabolizing phenotypes can be distinguished: normal metabolizers (NM) carry two normal function alleles (e.g., *CYP2C19*1/*1*); intermediate metabolizers (IM) carry one normal or one increased function allele and one no function allele (e.g., *CYP2C19*1/*2, CYP2C19*2/*17*); poor metabolizers (PM) carry two no function alleles (e.g., *CYP2C19*2/*3*); rapid metabolizers (RM) carry one normal and one increased function allele (i.e., *CYP2C19*1/*17)*, whereas carriers of two increased function alleles (i.e., *CYP2C19*17/*17*) are categorized as ultrarapid metabolizers (UM) [4,5].

Clopidogrel is the oldest and most extensively studied of the second-generation thienopyridines—platelet aggregation inhibitors of the P2Y12 receptor antagonist class. It is indicated for the prevention of occlusive arterial incidents (ischemic incidents) in patients at increased risk. As a monotherapy, or combined with aspirin, clopidogrel is indicated in patients who underwent percutaneous coronary intervention (PCI), embolization of cerebrovascular aneurisms or arterio-venous malformations (AVM), cerebrovascular stenting, in patients who have sustained acute myocardial infarction/acute coronary syndromes, transitory ischemic attack (TIA) or ischemic stroke, and in those with verified peripheral artery diseases (PAD) [6,7,8]. Regarding clopidogrel, CYP2C19 activity has a specific role: clopidogrel is a prodrug, and after oral administration, around 85% is inactivated in the first pass by the hepatic carboxylesterase 1 (CES1), whereas the remaining 15% requires enzymatic biotransformation into an active moiety, a reversible P2Y12 antagonist. While several hepatic CYP450 enzymes are involved, CYP2C19 is the most important one. This is a delicate role, since inadequate clopidogrel activation results in recurrent ischemic events, whereas excessive activation might, at least theoretically, increase the risk of bleeding [9,10,11,12,13,14]. The situation is further complicated by the fact that antiplatelet-treated patients commonly receive gastroprotective treatments, typically proton pump inhibitors, all of which are CYP2C19 substrates, and some (omeprazole, esomeprazole) are rather strong enzyme inhibitors [15]. The platelet response to clopidogrel shows considerable interindividual variability, and some 4–30% of patients fail to achieve adequate platelet inhibition [16,17]. After more than a decade of research, PM and IM CYP2C19 phenotypes have been extensively shown to result in lower active metabolite levels, and higher (preserved) platelet reactivity than NM (and, likely, RM and UM) phenotype, and are associated with an increased risk of major ischemic events (reviewed in, e.g., [4,12,18,19]). On the other hand, RM and UM patients display the highest active metabolite levels, but it is unclear whether this means greater efficacy and/or greater bleeding risk vs. NM patients—some studies have suggested that both could be the case [20,21,22,23,24], whereas others have suggested no association between the **17* allele and clopidogrel pharmacokinetics, pharmacodynamics, and ischemic or bleeding outcomes (i.e., no relevant difference between NM, RM, and UM subjects) after accounting for the **2* allele [25,26,27,28,29]. It has been postulated that the observed relationship between the **17* allele and clopidogrel antiplatelet effects may be due to absence of the no function **2* allele rather than presence of the increased function **17* allele [12,28]. The clinical utility of *CYP2C19* genotype-guided strategy for selection of P2Y12 inhibitors has been evaluated in multicenter randomized clinical trials [18]: PHARMCLO [30], POPular-Genetics [31], and TAILOR-PCI [32,33]. Overall, genotyping for the CYP2C19 no function alleles and identification of PM and IM subjects to guide P2Y12-targeted antiplatelet therapy has a reasonable clinical utility, whereas the identification of RM/UM subjects is not considered relevant [34].

Recently, a novel haplotype composed of two non-coding variants in the *CYP2C* gene cluster, *CYP2C18* NM_000772.3:c.*31T (rs2860840) and NM_000772.2:c.819+2182G (rs11188059), has been identified, and is reported to be associated with the ultrarapid metabolism of CYP2C19 substrates escitalopram and sertraline [35,36]. The haplotype is referred to as “*CYP2C:TG*”. It has been reported to be associated with exposure to and efficacy of a proton pump inhibitor omeprazole [37] and the azole antifungal voriconazole [38]. Another study found no association between the *CYP2C:TG* haplotype and in vivo exposure to CYP2C19 substrates citalopram, sertraline, voriconazole, omeprazole, pantoprazole, and rabeprazole in healthy volunteers, and no association between this haplotype and CYP2C18/2C19 abundance in human liver sample or activity in vitro [39].

Considering that the role of RM/UM phenotypes in bioactivation of clopidogrel is still doubtful, we aimed to explore whether accounting for the *CYP2C:TG* might add information about the risk of ischemic and bleeding events in patients treated with clopidogrel over 3–6 months. For this purpose, the incidence of events was analyzed with respect to the “classically” defined CYP2C19 phenotypes, that is, based on the *CYP2C19*2*17* genotypes, phenotypes defined with respect to the *CYP2C* cluster genotype, that is combined *CYP2C19*2*17* and *CYP2C* rs11188059 and rs2860840, and with respect to the presence of the *CYP2C:TG* haplotype.

## 2. Materials and Methods

### 2.1. Study Outline

The present data are preliminary results from a larger prospective study (“Pharmacogenomics in Prediction of Cardiovascular Drugs Adverse Reactions”) that started on 1 January 2022, and will last 60 months and include 1200 subjects. The study (ClinicalTrials.gov, NCT05307718) is conducted in accordance with the Declaration of Helsinki and was approved by the Ethics Committees of the School of Medicine, University of Zagreb, (reg. number 380-59-10106-20-111/125; class 641-01/20-02/01) and the University Hospital Centre Zagreb (class 8.1-20/142-2; number 02/21 AG), Croatia. All subjects provided written informed consent for genotyping the pharmacogenes of interest, and for publishing anonymized data for scientific purposes. The present analysis included consecutive adults (≥18 years of age), unrelated Caucasians of European (Slavic) descent, who started clopidogrel due to percutaneous coronary intervention (PCI), embolization of cerebrovascular aneurisms (AVM), or cerebrovascular stenting requiring >3 months of therapy, and were followed up over at least 3 and to a maximum of 6 months (all patients with no events were observed over 6 months) between 1 January 2022 and 1 March 2024. The decision to prescribe clopidogrel was at the discretion of attending cardiologist/stroke neurologist, and was in line with the approved prescribing information including recommendations to avoid it in patients who need to be treated with strong CYP2C19 inducers or inhibitors. Patients were genotyped for *CPY2C19*2*, *CYP2C19*17*, *CYP2C* rs11188059G>A, and rs2860840C>T. Phenotypes (PM, IM, NM, UM, RM) were determined based on the *CYP2C19*2*,**17* genotype, and on combined *CYP2C19*2,*17* and *CYP2C18* rs11188059 and *CYP2C18* rs2860840 genotypes (cluster genotype). They were also classified as *CYP2C:TG* carriers and non-carriers. Phenotypes/haplotype carriage were considered as “exposures” indicating genetically determined enzyme activity, and incidence of ischemic and bleeding events were recorded. Levels of exposure were then balanced on a number of relevant covariates, and their effects on outcomes of interest were estimated.

### 2.2. Patients

HIV patients were not included in this study. Patients undergoing PCI were not included if older than 80 years of age, were treated with continuous post-interventional glycoprotein IIb/IIIa inhibitors, had thrombocytopenia (<150 × 10^9^/L), more pronounced renal failure (creatinine > 200 µmol/L), anemia (hematocrit < 30%), hemorrhagic diathesis, history of major surgery within previous 6 weeks, or hemorrhagic stroke within previous 6 months. No particular exclusion criteria were implemented for neurovascular patients (i.e., undergoing cerebrovascular stenting or embolization of cerebrovascular aneurisms/arterio-venous malformations).

### 2.3. Genotyping Procedures

Genomic DNA was extracted from whole blood samples using the QIAamp DNA Blood Mini Kit (Qiagen, Hilden, Germany) according to the manufacturer’s protocol. Genotyping was performed using TaqMan^®^ Drug Metabolism Genotyping Assays or TaqMan^®^ SNP Genotyping Assays (Applied Biosystems, Carlsbad, CA, USA) for *CYP2C19*2* (c.681G>A, rs4244285; assay ID: C__25986767_70), *CYP2C19*17* (c.-806C>T, rs12248560; assay ID: C____469857_10), *CYP2C18 c.*31C>T* (rs2860840; assay ID: C__11201742_10) and *CYP2C18 c.819+2182G>A* (rs11188059; C__31983321_10), and TaqMan^®^Universal PCR Master Mix (Applied Biosystems, Carlsbad, CA, USA) by real-time PCR genotyping on the 7500 Real-Time PCR System (Applied Biosystems, Carlsbad, CA, USA), according to the manufacturer’s instructions. Genotyping of *CYP2C19*2* and **17* was performed with methods validated for routine pharmacogenetic testing in the clinical laboratory included in external quality assessment schemes, while rs11188059 and rs2860840 variants were validated by frequency for Caucasian population. Genotyping of *CYP2C19*2* and **17* was performed with methods implemented and validated for routine pharmacogenetic testing in the clinical laboratory that regularly participates in external quality assessment schemes (European Molecular Genetics Quality Network—EMQN; Reference Institute for Bioanalytics—RfB).

*CYP2C:TG* haplotype refers to the combination of T at *CYP2C18 c.*31C>T* (rs2860840) and G at *c.819+2182G>A* (rs11188059).

### 2.4. Genotype-Predicted Phenotypes and CYP2C:TG Haplotype

Phenotypes based on *CYP2C19*2,*17* genotype were defined as follows: (i) poor metabolizer (PM) = **2/*2*; (ii) intermediate metabolizer (IM) = **1/*2*, **2/*17*; (iii) normal metabolizer (NM) = **1/*1*; (iv) rapid metabolizer (RM) = **1/*17*; (v) ultrarapid metabolizer (UM) = **17/*17*. Phenotypes based on the *CYP2C19/CYP2C* cluster genotype were defined as follows: (i) PM = *CYP2C19 Null/CYP2C19 Null*; (ii) IM = *CYP2C19*17/CYP2C19 Null*, *CYP2C:TG/CYP2C19 Null* or *CYP2C19*1/CYP2C19 Null*; (iii) NM = *CYP2C19*1/CYP2C19*1*; (iv) RM = *CYP2C19*1/CYP2C19*17* or *CYP2C19*1/CYP2C:TG*; (v) UM = *CYP2C19*17/CYP2C19*17*, *CYP2C:TG/CYP2C:TG* or *CYP2C:TG/CYP2C19*17*. Finally, *CYP2C:TG* haplotype carriers were defined as those with *CYP2C* cluster diplotype TG/TG or “other”/TG (vs. “other”/“other”).

### 2.5. Outcomes of Interest

The following ischemic events were recorded: PCI target lesion, stent thrombosis, transitory ischemic attack (TIA), ischemic stroke (CVI), (re)hospitalization for myocardial ischemia, acute myocardial infarction/acute coronary syndrome, cardiovascular death, retinal ischemia, ischemia of the optic nerve, amaurosis fugax.

The following bleeding events were recorded: (i) all intracranial bleedings (e.g., nontraumatic intraparenchymal hematoma, subarachnoid, intraventricular, or subdural hemorrhage [40]), except for microhemorrhages and minor asymptomatic sulcal subarachnoid hemorrhages; (ii) any extracranial bleeding that required at least non-surgical medical intervention; (iii) any intraspinal or intraocular bleeding that compromised vision; (iv) fatal bleeding of any location or cause.

No grading based on severity of either ischemic or bleeding events was employed.

### 2.6. Data Analysis

All ischemic events and all bleeding events were considered jointly, regardless of severity or location. To estimate the effects of CYP2C19 phenotypes and of *CYP2C:TG* haplotype, different phenotypes (levels of exposure) and *CYP2C:TG* carriers and non-carriers were mutually balanced on several covariates that could have affected the outcome of interest: age, sex, indication for clopidogrel treatment (3 levels: percutaneous coronary intervention, embolism of cerebral aneurism, cerebrovascular stenting), chronic kidney, heart or liver failure (jointly), hypertension, diabetes or dyslipidemia, concurrent use of aspirin, concurrent use of anticoagulants, concurrent use of gastroprotective treatments (yes or no), and concurrent use of proton pump inhibitors (yes or no). For this purpose, we used entropy balancing [41] implemented in package WeightIt (Version 1.1.0) [42] in R [43]. Entropy balancing is a data preprocessing method that (where this is possible) guarantees covariate balance via a reweighting scheme that assigns a scalar weight to each sample unit such that the reweighted groups satisfy a set of balance constraints that are imposed on the sample moments of covariate distributions [41]. Standardized mean differences (d) < 0.1 between pairs of exposure (phenotype, haplotype) levels were considered to indicate adequate balance. Balanced data were analyzed by fitting weighted log-binomial models with robust variance estimation to binary outcomes to generate adjusted proportions. Since both the sample and number of events were limited, confidence intervals around relative risks would have been rather wide. Instead, we present adjusted proportions, their confidence intervals, and Cochran–Armitage test of trend in adjusted (weighted) proportions, where appropriate. We used SAS 9.4 for Windows (SAS Inc., Cary, NC, USA).

## 3. Results

### 3.1. Patient Characteristics

The present report addresses 283 patients (60.8% women) aged 22–85 years, who mostly (56.5%) underwent cerebral aneurism embolization and, less commonly, cerebrovascular stenting (27.9%) and PCI (15.6%) (Table 1). A history of ischemic cerebrovascular incidents (33.2%) and peripheral artery disease (38.2%) were common, and the most common comorbidities were hypertension (69.3%) and dyslipidemia (41.7%) (Table 1). Along clopidogrel, most patients were using aspirin (82.3%), whereas 13.1% were co-treated with anticoagulants (mainly direct oral anticoagulants) (Table 1). Gastroprotection was administered in 70.3% patients, typically proton pump inhibitors (mostly pantoprazole) (Table 1). The predominant phenotype based on the *CYP2C19*2,*17* genotype was “rapid metabolizer” (RM) (41.0%), followed by the “normal metabolizer” (NM) (31.1%) and “intermediate metabolizer” (IM) (18.7%). “Ultrarapid” (UM) and “poor metabolizer” (PM) phenotypes were rare/sporadic (6.7% and 2.5%, respectively) (Table 1). Most patients (73.8%) were wild-type homozygotes at *CYP2C* rs11188959G>A, whereas the prevalence of wild-type homozygotes (41.7%) and heterozygotes (48.1%) at *CYP2C* rs2860840C>T was similar (Table 1). There were 162 (57.2%) *CYP2C:TG* haplotype carriers (Table 1). When the phenotype was determined based on the combined **2*17*, rs11188059G>A, and rs2869840C>T genotypes, the prevalence of PM and IM remained unchanged, the prevalence of NM was greatly reduced to 23 (8.1%), and the prevalence of UM greatly increased to 95 (33.6%), while the prevalence of RM was 37.1% (Table 1). A total of 45 patients (15.9%) experienced ischemic events, and 49 (17.3%) experienced bleeding events, practically exclusively extracerebral (Table 1).

### 3.2. Outcomes across CYP2C19 Phenotypes—Raw Data

Although some phenotypes were uncommon (e.g., PM, UM by **2*,**17* classification, NM by cluster-based classification), a tendency of a numerically higher incidence of ischemic events and a lower incidence of bleeding events was apparent for PM/IM vs. NM, RM and UM phenotypes by either classification (Figure 1A,B). The tendency was more clearly visible when phenotype levels was collapsed to 3 (based on a biological rationale) (Figure 1C) for the combined PM/IM phenotype (*n* = 60), NM (*n* = 88 based on **2*,**17*, *n* = 23 based on the cluster-genotype), and RM/UM phenotypes (*n* = 135 based on **2*,**17*, *n* = 200 based on cluster-genotype) (Figure 1C). For the cluster-based phenotype, the tendency was apparent also when phenotypes were collapsed to PM/IM, combined NM and RM (*n* = 128), and UM (*n* = 95) (Figure 1C). This might be biologically more plausible, since a large prevalence of UM subjects in this phenotype classification is based predominantly on a “switch” of many of the NM and some of the RM subjects to the UM phenotype, indicating that there is a greater similarity between NM and RM subjects than between RM and UM subjects.

### 3.3. Outcomes across Phenotypes—Balanced Data

Patients across the collapsed levels of CYP2C19 phenotypes (PM/IM combined, NM, and RM/UM combined based on **2*,**17*; and PM/IM, NM, and RM combined, and UM based on the *CYP2C19/CYP2C* cluster) differed largely with respect to most of the covariates (Table 2).

After entropy balancing, all subsets were mutually closely similarly with an adequate balance (all d < 0.1) (Table 3). The adjusted proportions showed similar trends as the raw data: numerically higher proportion of ischemic events and lower proportion of bleeding events in PM/IM patients vs. NM and combined RM/UM patients, or vs. the combined NM/RM and UM patients (Figure 2).

The sample was modest in size and the number of events was low, so formal tests of trend in proportions for all outcomes yielded relatively low z-values. However, despite this lack of precision of the estimates, numerical trends appeared obvious and were similar for both types of phenotype definitions (Figure 2).

### 3.4. Outcomes Regarding the CYP2C:TG Haplotype

The *CYP2C:TG* carriers (*n* = 162) and non-carriers (*n* = 121) mildly differed regarding the covariates, and the prevalence of ischemic events (14.8% vs. 17.4%) was generally similar between them (Table 4). The prevalence of bleeding events appeared to be slightly lower in haplotype carriers (14.8% vs. 20.7%, d = −0.153). After entropy balancing, virtually identical values for all covariates were achieved for haplotype carriers and non-carriers: the incidence of ischemic events remained closely similar (14.1% vs. 16.1%) in *CYP2C:TG* carriers and non-carriers, whereas the difference in bleeding events was minimal (14.9% vs. 20.1%) (Table 4).

## 4. Discussion

As a result of extensive investigations—including genome-wide association studies and candidate-gene analyses of observational data and data from large randomized trials—the current guidelines by the Clinical Pharmacogenetic Implementation Consortium on the *CYP2C19* genotype (genotype-predicted phenotype) in clopidogrel-treated patients *strongly* recommend that in cardiovascular indications, clopidogrel should be avoided in PM/IM CYP2C19 metabolizers (preference towards the non-CYP2C19-dependent P2Y12 antagonists like prasugrel or ticagrelor) [4]. A *moderate* recommendation for clopidogrel avoidance in PM/IM patients refers to cerebrovascular indications [4]. No recommendation pertains to RM or UM patients in either indication, implying that differentiation between NM, RM, and UM subjects appears to be of no clinical utility [4]. The considerable inter-subject variability in response to clopidogrel is considered largely genetically determined [44], and only partly explained by identification of the loss-of-function *CYP2C19* alleles [45,46,47].

Polymorphisms in several other genes (e.g., *CYP2B6*, *CES1*, *SCOS5P1*, *CDC42BPA*, *CTRAC1*, *ABCB1*, *G4GALT2*, *P2RY12*, *PON1*) have been suggested as relevant for the antiplatelet effects of clopidogrel, although their impact on “hard” clinical outcomes is yet to be clarified [27,28,48,49]. It is reasonable to envisage a risk-grading system encompassing multiple genetic variants and non-genetic factors that would be more predictive and reliable than any individual indicator alone, regarding the clinical efficacy/safety of clopidogrel in cardiovascular and neurovascular indications [27,50].

The newly discovered *CYP2C:TG* haplotype has been shown associated with accelerated metabolism (by around 25%) of CYP2C19 substrates escitalopram and sertraline [35,36], and it has been suggested that it could affect the exposure to/effects of other CYP2C19 substrates such as omeprazole [37] and voriconazole [38]. However, opposing observations (not indicating effects of *CYP2C:TG*) have been reported, as well [39]. Currently, it is considered that the *CYP2C:TG* haplotype provides useful additional information to predict enhanced CYP2C19 activity, an effect that might be ethnically specific, but further investigations are needed before it could be routinely used in clinical settings [51].

Considering that in the case of CYP2C19 phenotypes and clopidogrel efficacy/safety, “rapid” or “ultrarapid” designation has not been thus far considered practically relevant (i.e., no distinction has been made vs. normal metabolizers), we considered it reasonable to explore a concept that “additional information” about enhanced activity conveyed by the *CYP2C:TG* haplotype might also additionally inform about efficacy and bleeding risks in clopidogrel-treated patients. For this purpose, we assessed the incidence of ischemic and bleeding events with respect to PM/IM, NM, and RM/UM phenotype based on the “classical” criteria of the *CYP2C19*2*17* genotype, PM/IM, NM/RM, and UM phenotype informed by the combined *CYP2C19*2*17* genotype and rs2860840 and rs11188059 (cluster *CYP2C19/CYP2C* genotype), and with respect to the presence of the *CYP2C:TG* haplotype. Due to the limited sample size and limited number of events, the present risk estimates are imprecise (wide confidence intervals), but numerical trends are obvious: higher incidence of ischemic events in PM/IM patients vs. mutually similar NM/RM/UM patients by either classification; lower incidence of bleeding events in PM/IM patients vs. NM/RM/UM (also mutually similar) by either classification; a similar incidence of ischemic events in *CYP2C:TG* carriers and non-carriers, and a mildly lower incidence of bleeding events in the former than in the latter. Despite this imprecision, however, the patient subsets were not extremely small (between 60 and 135 across the collapsed phenotype categories, 162 vs. 121 regarding *CYP2C:TG*), and the observed proportions do not appear critically fragile: indeed, one or two events more or less could have occurred in any subset by shear chance, but it is unlikely that this would have substantially changed the observed numerical trends. It could be objected that mixing cardiovascular and neurovascular patients/indications was not appropriate, but we deemed it a feasible option since clopidogrel is indicated in both settings, where it is expected to convey similar effects on ischemic/bleeding events through the same mechanism of action. Also, some residual confounding should be attributed to the fact that we did not consider factors that could have influenced effects of the commonly co-administered aspirin, or other already mentioned genetic factors that have been suggested to affect the antiplatelet effect of clopidogrel. On the other hand, we did account for a number of factors known to have an impact on the risk of ischemic or bleeding events in clopidogrel-treated patients: (i) in line with the prescribing recommendations, clopidogrel was avoided in patients using strong CYP2C19 inducers or inhibitors, and in HIV-positive patients; (ii) exposed and control subjects were balanced according to age, chronic kidney disease, hypertension, dyslipidemia, diabetes, serious liver disease, known history of coronary, cerebral or peripheral artery disease (and, by virtue of this, also indirectly regarding treatments used in such patients); (iii) concomitant aspirin or anticoagulant use [27,50], the use of gastroprotection (which may directly reflect on incidence of gastro-intestinal bleeding), and, specifically, the use of proton pump inhibitors (PPIs). This latter factor is of two-fold relevance: (i) all PPIs are CYP2C19 substrates that could compete with clopidogrel or inhibit it [15] and thus affect clopidogrel bioactivation and its effects [52,53]. Of note, the effects of pantoprazole and rabeprazole used in the present sample of patients are minimal, if any, both in vitro and in vivo [15,52,53]; (ii) *CYP2C:TG* could affect their clearance/exposure and, thus, efficacy in gastroprotection. In this regard, it is of note that no gastro-intestinal bleedings were recorded during the observed period. By using the entropy balancing method, all contrasted “exposures” (different phenotypes, haplotypes) were perfectly balanced with respect to all these factors. Finally, certain validity to the present estimates is conveyed by the fact that In PM/IM, NM, and RM/UM patients classified based on “classical” criteria of the *CYP2C19*2*17* genotype, we observed the incidence of ischemic and bleeding events in line with the theoretical expectations [4], and the observations based on the *CYP2C19/CYP2C* cluster genotype-based phenotypes were closely similar. Under these circumstances, it appears justified to state that no clear-cut, strong signal has been detected in the present exploration that would indicate a major role of the *CYP2C:TG* haplotype for informing the risk of treatment failure or bleeding events in clopidogrel-treated patients.

## 5. Conclusions

In the present analysis, we explored the possibility that the awareness about the newly discovered haplotype *CYP2C:TG* might convey useful information to anticipate the risk of treatment failure (ischemic events) or the risk of bleeding in clopidogrel-treated Caucasians of European (Slavic) descent. With the limitations of a modest sample size, and possible residual confounding, we failed to observe clear-cut strong signals to support such possibilities. However, having in mind racial/ethnic differences in patient responsiveness to P2Y12 inhibitors in general, and specifically to clopidogrel [54], the situation in other populations might be substantially different.

## Figures and Tables

**Figure 1 genes-15-00607-f001:**
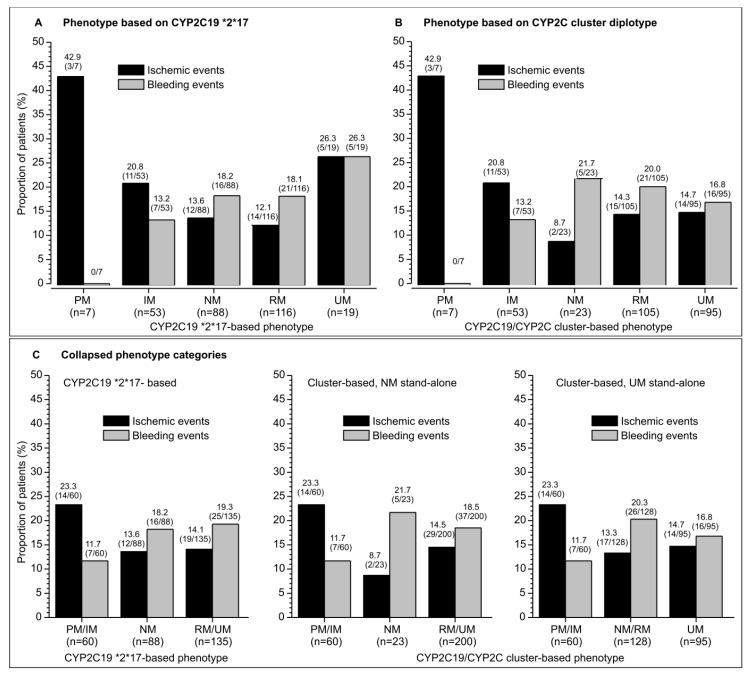
Raw incidence of ischemic and bleeding events in clopidogrel-treated patients with different CYP2C19 phenotypes: poor (PM), intermediate (IM), normal (NM), rapid (RM), and ultrarapid (UM) metabolizer. Depicted are percentages and number of patients with events/total number in brackets. (**A**) Incidence of events across phenotypes defined based on the *CPY2C19*2,*17* genotype. (**B**) Incidence of events across phenotypes defined based on the *CYP2C19/CYP2C* cluster genotype. (**C**) Incidence of events across collapsed phenotype categories.

**Figure 2 genes-15-00607-f002:**
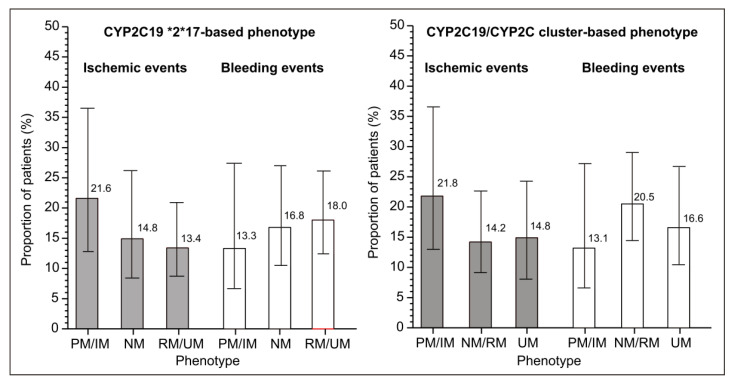
Adjusted incidence (based on the fully balanced data in Table 3) of ischemic and bleeding events in clopidogrel-treated patients with different CYP2C19 phenotypes: poor (PM), intermediate (IM), normal (NM), rapid (RM), or ultrarapid (UM) metabolizer. Depicted are percentages from the weighted pseudopopulations.

**Table 1 genes-15-00607-t001:** Patient characteristics. Values are *n* (percent) or mean ± SD (minimum–maximum) for age.

Characteristic	Value	Characteristic	Value
N	283	*CYP2C19*2*17* phenotype	
Age (years)	60 ± 12 (22–85)	Normal (NM)	88 (31.1)
Female/male	172 (60.8)/111	Intermediate (IM)	53 (18.7)
Indication		Poor (PM)	7 (2.5)
Percutaneous coronary intervention	44 (15.6)	Rapid (RM)	116 (41.0)
Cerebral aneurism embolization	160 (56.5)	Ultrarapid (UM)	19 (6.7)
Cerebrovascular stenting	79 (27.9)	*CYP2C* rs11188059G>A	
Medical history/comorbidity		GG	209 (73.8)
Chronic kidney disease	19 (6.7)	GA	71 (25.1)
Chronic heart failure	16 (5.6)	AA	3 (1.1)
Chronic liver failure/cirrhosis/NAFLD	5 (1.8)	*CYP2C* rs2860840C>T	
Diabetes mellitus	43 (15.2)	CC	118 (41.7)
Hypertension	196 (69.3)	CT	136 (48.1)
History of ischemic stroke/TIA	94 (33.2)	TT	29 (10.2)
Peripheral artery stenosis	108 (38.2)	*CYP2C* cluster diplotype	
Coronary artery disease	25 (8.8)	CG/CG	111 (39.2)
Dyslipidemia	118 (41.7)	CG/TG	89 (31.5)
Concomitant anticoagulant, total	37 (13.1)	CA/TG	46 (16.3)
Direct oral anticoagulant	33 (11.7)	TA/TG	18 (6.4)
Warfarin	3 (1.1)	TG/TG	9 (3.2)
Enoxaparin	1 (0.4)	CA/CG	7 (2.5)
Concomitant aspirin	233 (82.3)	TA/TA	2 (0.7)
Gastroprotective treatment, total	199 (70.3)	CA/TA	1 (0.4)
Pantoprazole	168 (59.4)	*CYP2C:TG* carrier	162 (57.2)
Esomeprazole	6 (2.1)	*CYP2C19/2C* phenotype	
Ranitidine	24 (8.5)	Normal (NM)	23 (8.1)
Rabeprazole	1 (0.4)	Intermediate (IM)	53 (18.7)
*CYP2C19*2*		Poor (PM)	7 (2.5)
**1/*1*	223 (78.8)	Rapid (RM)	105 (37.1)
**1/*2*	53 (18.7)	Ultrarapid (UM)	95 (33.6)
**2/*2*	7 (2.5)	Ischemic events total	45 (15.9)
*CYP2C19*7*		CVI, retinal/opticus/stent	40 (14.1)
**1/*1*	131 (46.3)	TIA or amaurosis fugax	5 (1.8)
**1/*17*	133 (47.0)	Bleeding events total	49 (17.3)
**17/*17*	19 (6.7)	Cerebral	6 (2.1)
		Extracerebral ^a^	44 (15.5)

^a^ Pseudoaneurism, large local hematoma, cutaneous/subcutaneous hematoma, and other. CVI—cardiovascular insult; NAFLD—non-alcoholic fatty liver disease; TIA—transitory ischemic attack.

**Table 2 genes-15-00607-t002:** Patient characteristics across the CYP2C19 phenotypes (poor, intermediate, normal, rapid, ultrarapid metabolizers [PM, IM, NM, RM, UM, respectively]) defined based on the *CYP2C19*2/*17* genotype or based on the *CPY2C19/CYP2C* cluster genotype. Data are mean ± SD or count (percent). Shown are maximum standardize mean differences (Max d) between any two of the three phenotype levels.

	Phenotypes Based on *CYP2C19*2*17* Genotype				Phenotypes Based on *CYP2C19/CYP2C* Cluster Genotype			
	PM/IM Phenotype	NM Phenotype	RM/UM Phenotype	Max d	PM/IM Phenotype	NM/RM Phenotype	UM Phenotype	Max d
N	60	88	135	---	60	128	95	---
Age (years)	61 ± 13	59 ± 12	61 ± 12	0.112	61 ± 13	60 ± 12	61 ± 13	0.116
Sex								
Female	26 (43.3)	52 (59.1)	94 (69.6)	0.545	26 (43.3)	81 (63.3)	65 (68.4)	0.522
Male	34 (56.7)	36 (40.9)	41 (30.4)	0.545	34 (56.7)	47 (36.7)	30 (31.6)	0.522
Indication								
PCI	12 (20.0)	14 (15.9)	18 (13.3)	0.181	12 (20.0)	18 (14.1)	14 (14.7)	0.161
Brain AVM	29 (48.3)	47 (53.4)	84 (62.2)	0.281	29 (48.3)	70 (54.7)	61 (64.2)	0.322
CV stenting	19 (31.7)	27 (30.7)	33 (24.4)	0.159	19 (31.7)	40 (31.2)	20 (21.1)	0.238
CKD/CHF/CLD	4 (6.7)	16 (18.2)	10 (7.4)	0.377	4 (6.7)	18 (14.1)	8 (8.4)	0.251
Hypertension	40 (66.7)	57 (64.8)	99 (73.3)	0.184	40 (66.7)	85 (66.4)	71 (74.7)	0.181
DM/dyslipidemia	30 (50.0)	39 (44.3)	67 (49.6)	0.114	30 (50.0)	58 (45.3)	48 (50.5)	0.104
CVI/PAD/CAD	28 (46.7)	44 (50.0)	67 (49.6)	0.067	28 (46.7)	65 (50.8)	46 (48.4)	0.082
Gastroprotection	40 (66.7)	60 (68.2)	99 (73.3)	0.145	40 (66.7)	91 (71.1)	68 (71.6)	0.107
Use PPI	36 (60.0)	49 (55.7)	90 (66.7)	0.226	36 (60.0)	79 (61.7)	60 (63.2)	0.065
Use aspirin	44 (73.3)	79 (90.0)	110 (81.5)	0.430	44 (73.3)	115 (89.8)	74 (77.9)	0.422
Use anticoagulants	9 (15.0)	8 (9.1)	20 (14.8)	0.177	9 (15.0)	12 (9.4)	16 (16.8)	0.218

AVM—aneurisms of arteriovenous malformations; CAD—coronary artery disease; CHF—chronic heart failure; CKD—chronic kidney failure; CLD—chronic liver failure; CV—cerebrovascular; CVI—cerebrovascular incident; PAD—peripheral artery disease; PCI—percutaneous coronary intervention; PPI—proton pump inhibitor.

**Table 3 genes-15-00607-t003:** Patient characteristics across the CYP2C19 phenotypes (PM, IM, NM, RM, UM) defined based on the *CYP2C19*2*17* genotype or on the *CPY2C19/CYP2C* cluster genotype after entropy balancing ^1^. Data are mean ± SD or percent patients with a characteristic in the weighted pseudopopulation. Shown are maximum standardize mean differences (Max d) between any two of the three phenotype levels.

	Phenotypes Based on *CYP2C19*2*17* Genotype ^1^				Phenotypes Based on *CYP2C19/CYP2C* Cluster Genotype ^2^			
	PM/IM Phenotype	NM Phenotype	RM/UM Phenotype	Max d	PM/IM Phenotype	NM/RM Phenotype	UM Phenotype	Max d
N	60	88	135	---	60	128	95	---
Age (years)	60 ± 12	60 ± 12	60 ± 12	0.017	60 ± 12	60 ± 12	60 ± 12	0.016
Sex								
Female	58.2	61.7	61.4	0.072	57.9	61.2	61.7	0.078
Male	41.8	38.3	38.6	0.072	42.1	38.8	38.3	0.078
Indication								
PCI	16.7	15.8	15.4	0.035	16.9	15.5	15.4	0.040
Brain AVM	54.8	57.2	57.2	0.049	54.5	56.6	57.4	0.059
CV stenting	28.5	27.0	27.4	0.034	28.6	27.9	27.2	0.032
CKD/CHF/CLD	9.3	10.9	10.0	0.054	9.3	10.7	10.1	0.049
Hypertension	67.9	68.6	69.3	0.031	68.0	69.3	69.6	0.035
DM/dyslipidemia	47.0	47.2	47.7	0.012	47.1	48.0	47.6	0.017
CVI/PAD/CAD	48.0	48.4	49.4	0.029	47.8	49.0	49.5	0.033
Gastroprotection	69.5	69.9	70.9	0.033	69.4	70.3	71.0	0.035
Use PPI	60.7	61.2	62.4	0.034	60.8	61.8	62.4	0.034
Use aspirin	80.8	83.8	82.1	0.079	80.5	83.1	81.9	0.067
Use anticoagulants	13.6	12.6	13.2	0.032	13.6	13.0	13.4	0.019

^1^ Effective sample sizes were 49.3 for PM/IM, 75.5 for NM, and 123 for RM/UM. Entropies by phenotype level were 0.108, 0.082, and 0.048, respectively. Weights ranged from 0.382 to 1.847 in all groups, with coefficients of variation of 0.470, 0.410, and 0.315, respectively. ^2^ Effective sample sizes were 49.6 for PM/IM, 117 for NM/RM, and 85.5 for UM. Entropies by phenotype level were 0.102, 0.046, and 0.052, respectively. Weights ranged from 0.486 to 1.903 in all groups, with coefficients of variation of 0.462, 0.314, and 0.335, respectively. AVM—aneurisms of arteriovenous malformations; CAD—coronary artery disease; CHF—chronic heart failure; CKD—chronic kidney failure; CLD—chronic liver failure; CV—cerebrovascular; CVI—cerebrovascular incident; PAD—peripheral artery disease; PCI—percutaneous coronary intervention; PPI—proton pump inhibitor.

**Table 4 genes-15-00607-t004:** Patient characteristics in *CYP2C:TG* carriers (TG-other, TG-TG) and non-carriers in the *CYP2C* cluster diplotype and ischemic and bleeding events before and after entropy balancing. Data are mean ± SD for age, and count (percent) for raw data, or percent in the entropy-balanced data ^1^. Depicted are standardized mean differences (d; values < 0.1 indicate fully adequate balance). Shown are also outcomes of interest, i.e., incidence of ischemic and bleeding events: for raw data (before entropy balancing of covariates), simple proportions are provided; for adjusted (balanced) data, incidence is given with 95% confidence intervals.

	Before Entropy Balancing			After Entropy Balancing		
	TG Carrier	TG Non-Carrier	d	TG Carrier	TG Non-Carrier	d
N	162	121	---			---
Age (years)	59 ± 13	62 ± 12	−0.265	60 ± 12	60 ± 13	−0.025
Sex						
Female	100 (61.7)	72 (59.5)	0.045	61.0	60.7	0.005
Male	62 (38.3)	49 (40.5)	−0.045	39.0	39.3	−0.005
Indication						
PCI	23 (14.2)	21 (17.4)	−0.087	15.4	16.0	−0.015
Brain AVM	97 (59.9)	84 (52.1)	0.158	56.8	56.1	0.015
CV stenting	42 (25.9)	121 (30.6)	−0.103	27.7	27.9	−0.003
CKD/CHF/CLD	19 (11.7)	11 (9.1)	0.086	10.7	10.4	0.010
Hypertension	108 (66.7)	88 (72.7)	−0.132	69.3	69.6	−0.005
DM/dyslipidemia	76 (46.9)	60 (49.6)	−0.054	48.0	48.3	−0.005
CVI/PAD/CAD	79 (48.8)	60 (49.6)	−0.016	49.2	49.1	0.002
Gastroprotection	114 (70.4)	85 (70.2)	0.003	70.3	70.3	−0.000
Use PPI	99 (61.1)	76 (62.8)	−0.035	61.8	61.9	−0.003
Use aspirin	132 (81.5)	101 (83.5)	−0.052	82.2	82.5	−0.007
Use anticoagulants	23 (14.2)	14 (11.6)	0.078	13.2	12.8	0.012
*Outcomes*						
Ischemic events	24 (14.8)	21 (17.4)	−0.069	14.1 (9.7–20.7)	16.1 (10.7–24.2)	−0.056
Bleeding events	24 (14.8)	25 (20.7)	−0.154	14.9 (10.2–21.7)	20.1 (14.1–28.8)	−0.128

^1^ Effective sample sizes were 156 for TG carriers and 116 for non-carriers. Entropies were 0.018 and 0.021, respectively. Weights ranged from 0.694 to 1.375 in both groups, with coefficients of variation of 0.189 and 0.208, respectively. AVM—aneurisms of arteriovenous malformations; CAD—coronary artery disease; CHF—chronic heart failure; CKD—chronic kidney failure; CLD—chronic liver failure; CV—cerebrovascular; CVI—cerebrovascular incident; PAD—peripheral artery disease; PCI—percutaneous coronary intervention; PPI—proton pump inhibitor.

## Data Availability

The datasets generated during and/or analyzed during the current study are available from the corresponding author on reasonable request.

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
