# Peer review of "Is CYP2C Haplotype Relevant for Efficacy and Bleeding Risk in Clopidogrel-Treated Patients?"

_genes, 2024, doi:10.3390/genes15050607_

Round 1

Reviewer 1 Report

Comments and Suggestions for Authors

Comments:

1. The text of lines 73 to 75 is confusing, is there greater efficacy or greater risk of bleeding, or both?

2. Although genotypes are associated with the type of metabolism, as stated in your introduction, some drugs may affect the pharmacokinetics of clopidrogel, why was drug level monitoring not performed?

3. Regardless of the type of drug (anticoagulant, proton pump inhibitor, etc.), it would be relevant to introduce as covariates whether the patient was concomitantly receiving a) a Cyp2C19 substrate, b) a Cyp2C19 inhibitor.

4. In lines 204-224, "table 1" is cited more than 10 times, I suggest that you delete some of them and write at the end of the paragraph "all the above data are shown in Table 1".

5. In Table 3, why are the age and SD data the same in all columns? Is this a mathematical adjustment?

6. In section "3.4. Outcomes regarding the CYP2C:TG haplotype" Table 4 is cited four times. Similar to the comment for Table 1, decrease the number of times this Table is mentioned.

7.  The Discussion section should be restructured and should contrast the results of the study with those published in the literature ("pro" or "con"), instead of presenting a justification for the treatment of the data, which could be moved to the Results section.

8. A Conclusions section should be included to present the findings of the study.

The Contributions, Funding, IRB, Informed Consent, and Data Availability sections are missing.

Minor coments:

Use British or American English, e.g.: categorized (line 39) or categorised (line 48), embolisation/embolization, etc).

Single-use abbreviations are unnecessary (Pharmvar, GWAS, CPIC, etc).

PM and IM abbreviations have already been described above (line 69).

Line 73 is missing the closing parenthesis.

Some misspelled words or typos in the text are underlined.

The meaning of PM, IM is reiterated, indicating it in the first occurrence would be enough. 

Table 3 must be adjusted.

Table 4 must be adjusted.

Author Response

Comment 1. The text of lines 73 to 75 is confusing, is there greater efficacy or greater risk of bleeding, or both?

Reply. We agree. The part of the sentence was revised to read: “….some studies have suggested that both could be the case….”.

Comment 2. Although genotypes are associated with the type of metabolism, as stated in your introduction, some drugs may affect the pharmacokinetics of clopidrogel, why was drug level monitoring not performed?

Reply. We are not sure that we fully understand the question. If it aimed to ask why clopidogrel blood concentrations were not measured, we reply that several tests have been devised over the years that aimed to quantify the phenomenom of “platelet reactivity” (residual to exposure to antiplatelet compounds) as aids in optimizing antiplatelet treatment. Several large trials have been conducted in order to estimate whether such “guided” therapy was likely to confer benefits in terms of improved efficacy and/or safety, but have failed to clearly demonstrate either. In other words, available evidence does not support the use of laboratory tests to guide the dosing of clopidogrel  (see eg. overviews in Front Cardiovasc Med 2022; 9: https://doi.org/10.3389/fcvm.2022.805525; Blood 2017; 130:713-721; Pharmacol Res Perspect 2020; 8:e00686). Monitoring clopidogrel plasma concentrations (or metabolites) has never been established as a routine valuable procedure. Hence, it is not regularly practiced. This was an observational study that enrolled “real-life” patients with the intention to explore whether “additional pharmacogenetic knowledge” might confer major useful information to assess the risks of ischemic events and/or bleeding; hence, none of the patients was monitored for clopidogrel (and active metabolite) concentrations. We do not see that this is a limitation of the present study.

Comment 3. Regardless of the type of drug (anticoagulant, proton pump inhibitor, etc.), it would be relevant to introduce as covariates whether the patient was concomitantly receiving a) a Cyp2C19 substrate, b) a Cyp2C19 inhibitor.

Reply. If one consults sources that are bound (e.g., by regulatory requirements) to provide detailed and updated information about any proven or theoretically plausible or possible interaction between clopidogrel and other drugs (for example, a recently updated Summary of Product Characteristics at the Electronic Medicines Compendium – available at Plavix 300 mg film-coated tablets - Summary of Product Characteristics (SmPC) - (emc) (medicines.org.uk), accessed May 1, 2024), it might seem impossible to conduct a reasonably valid study (experimental/randomized or observational) addressing elements that might contribute to comprehension of factors informative about occlusive events/bleeding risks associated with clopidogrel use, that would be smaller than several thousands of patients: the list of such possible concomitant drugs is extremely long. However, the prescribing recommendations are rather straightforward about clinically relevant outcomes: i) bleeding risk is affected by concomitantly used other platelet inhibitors and anticoagulants, which is commonly therapeutically needed. This was taken into account; ii) there are medical conditions that by per se increase the risk of bleeding, but also require antiplatelet treatment. This was taken into account; iii) it is strongly advised that clopidogrel is avoided in patients who need to be treated with strong CYP2C19 inducers or inhibitors (like rifampicin, most of the azole antifungals, classical antiseizure medications like phenytoin, carbamazepine, phenoparbital, primidone, antidepressants fluvoxamine, fluoxetine); iv) antiretroviral boosted treatments are known to be associated with increased risk of bleeding in clopidogrel treated patients- and the recommendations is that these are avoided.Patients in this cohort were treated with clopidogrel in line with the presecribing recommendations – thus, possible effects of the listed (and related) drugs are controlled. Moreover, by balancing patients on relevant comorbidities, additional adjustment (although not explicitly stated) is indirectly achieved regarding treatments for these conditions, at least to some extent. As for PPIs – it is nowadays well known that omeprazole and esomeprazole are of primary interest as sources of potential interaction with clopidogrel. These were not used in this cohort. On the other hand, gastro-protection is commonly required to prevent GI bleedings. In fact, if one were to carefully consider prescribing recommendations, it would come to the fact that PPIs were virtually unavoidable, and if one were to follow other recommendations, it would come to the fact that these are potentially the major source of biotransformation interactions with clopidogrel. Concomitant use of CYP2C19 substrates is not considered a soruce of practically relevant interactions. Hence, we adjusted for (i) use of gastroprotection and (ii) specifically for PPI use (PPIs for which potential for interaction with clopidogrel is considered minimal). Therefore, some points were not explicitly stated in the submitted manuscript (for example, that clopidogrel was used in line with the approved prescribeing recommendations), or that we did not include HIV-positive patients (but, anyhow, clopidogrel would not have been prescribed), and this might introduce doubts to potential readers. Consequently, we introduced the following revisions: i) in the Material and Methods section, subsection 2.1. Study outline, p. 3, we extended the sentence starting in Line 122 to read “The decision to prescribe clopidogrel was at a discreation of the attending cardiologist/stroke neurologist, and was in line with the approved prescribing information including recommendations to avoid it in patients who need to be treated with strong CYP2C19 inducers or inhibitors”; ii) in the Materials and Methods section, subsection 2.2 Patients, p.3, (new line 134) we added an explicit statement that HIV positive patients were not included in the study; iii) we revised a part of the Discussion section between lines 395 and 402, p.12, to sum-up the factors that might have affected the outcomes, but were accounted for by different means (avoidance of clopidogrel concomitantly with strong CYP2C19 inhibitors or inducers and in HIV positive patients; covariate balancing). With respect to this point about confounding – the few previous reports dealing with the haplotype in question (none related to clopidogrel) reported exclusively unadjusted data.

Comment 4. In lines 204-224, "table 1" is cited more than 10 times, I suggest that you delete some of them and write at the end of the paragraph "all the above data are shown in Table 1".

Reply. We intentionally repeatedly referred to the Table (as we did with other tables) – so that a potential readers is not confused about finding the data that we are referring to. We tried to reduce the number of table citations.

Comment 5. In Table 3, why are the age and SD data the same in all columns? Is this a mathematical adjustment?

Reply. Yes – this is clearly stated in the Table title and the text – this is balance achieved by entropy balancing.

Comment 6. In section "3.4. Outcomes regarding the CYP2C:TG haplotype" Table 4 is cited four times. Similar to the comment for Table 1, decrease the number of times this Table is mentioned.

Reply. We intentionally repeatedly referred to the Table (as we did with other tables) – so that a potential readers is not confused about finding the data that we are referring to. We tried to reduce the number of table citations.

Comment 7. The Discussion section should be restructured and should contrast the results of the study with those published in the literature ("pro" or "con"), instead of presenting a justification for the treatment of the data, which could be moved to the Results section.

Reply. The currently existing data pertinent to the investigated haplotype are very limited, all are explicitly presented in the Introduction and Discussion sections. Please note – none of the existing data refers specifically to the investigated haplotype and clopidogrel. Hence, there is no data to which the present findings could be compared/discussed. On the other hand, we considered it important to discuss the methodological aspects of the present study, because perception of the presented data as reasonably reliable critically depends on the employed methodology. No revisions were introduced.

Comment 8. A Conclusions section should be included to present the findings of the study.

Reply. The Journal offers a possibility, but does not require that a manuscript contains a “Conclusion” section. In the light of some of this comment and a comment by the 3rd reviewer (to explicitly state racial/ethnic “exclusivity” of the present data), we added a Conclusions section. Related to it, we added an additional reference (ref. 54).

Comment 9. The Contributions, Funding, IRB, Informed Consent, and Data Availability sections are missing.

Reply. IRB and Informed consent are explicitly listed at p. 3, Methods section, subsection 2.1 Study outline. Funding statement is clearly listed at the end of the manuscript, after Discussion, before the references (as “acknowledgments”). We introduced the following revisions: i) we changed “acknowledgments” to “Funding”; ii) we added “Author contributions” and “Data availability”.

Comment 10. Use British or American English, e.g.: categorized (line 39) or categorised (line 48), embolisation/embolization, etc).

Reply. Corrected.

Comment 11. Single-use abbreviations are unnecessary (Pharmvar, GWAS, CPIC, etc).

Reply. Corrected.

Comment 12. PM and IM abbreviations have already been described above (line 69).

Reply. We intentionally commonly repeated full names of metabolizers phenotypes (toghether with abbreviations) – since PM, RM, UM etc, after a certain segment of text might again seem not straightforward to understand to readers not commonly dealing with this topic. However, we tried to reduce this in the revised text.

Comment 12. Line 73 is missing the closing parenthesis.

Reply. Thank you for spotting it - corrected.

Comment 13. Some misspelled words or typos in the text are underlined.

Reply. Thank you. We re-checked spelling.

Comment 14. The meaning of PM, IM is reiterated, indicating it in the first occurrence would be enough.

Reply. We intentionally commonly repeated full names of metabolizer phenotypes (toghether with abbreviations) – since PM, RM, UM etc, after a certain segment of text might not seem straightforward to understand to readers not commonly dealing with this topic. However, we tried to reduce this in the revised text.

Comment 15. Table 3 must be adjusted. Table 4 must be adjusted.

Reply. Done.

Reviewer 2 Report

Comments and Suggestions for Authors

This study investigated the drug response of clodipogrel in patients with different noble CYP2C haplotypes and genotypes of CYP2C19. I have some comments that I believe might help the authors in increasing the impact of this manuscript.

1. It would be nice to do some statistical analysis on the key observations.

2. Line 22-23, line 101, line 219, Table and Figure: I am confused as to whether you are referring to CYP2C19*2/*17 genotype or subjects with the CYP2C19*2 or CYP2C19*17 allele. The author should clarify the wording as it may be confusing to the reader.

3. Line 36: In my opinion, it would be better to replace it with “The cytochrome P-450 isozymes 2C19 (CYP2C19)~”.

4. line 37: Replace 35 star to 39 star.

Author Response

Comment 1. It would be nice to do some statistical analysis on the key observations.

Reply. Please note – we did “statistically analyze” the data: i) data were first pre-processed,i.e., entropy balancing was used to achieve conditional exchangeability between exposed and controls (i.e., balance on measured covariates); ii) then, weighted regression models were fitted to outcomes. The results are presented as adjusted proportions. We avoided reporting their ratios (i.e., relative risks, RR), because, as it should be clearly understandable from the shown results, these ratios would have had very wide confidence intervals. For example, in Figure 2, a ratio of a proportion 21.6% with 95%CIs extending from 13 to 36.5% over a proportion of e.g., 13.4%, with CIs extending from 8 tp 22% is 1.61, but CIs are very wide. We deemed that for the illustration of the major point, it would suffice to show numerical trends, i.e., trends in proportions. We did also test for the (linear) trends in proportions, but we deemd that reporting relatively low z-values with corresponding P-values for trends of 0.07 or 0.09 or 0.120 etc. – would not convey any relevant information. Moreover, this could be confusing for readers not fully acquainted with the meaning and interpretation of P-values. We therefore focused on illustration of numerical trends which are in line with theoretical expectations – and the fact that by accounting for the investigated haplotype we observed no clear-cut strong signal that it mattered regarding these trends.

Comment 2. Line 22-23, line 101, line 219, Table and Figure: I am confused as to whether you are referring to CYP2C19*2/*17 genotype or subjects with the CYP2C19*2 or CYP2C19*17 allele. The author should clarify the wording as it may be confusing to the reader.

Reply. We used the recommended nomenclature: CYP2C19*2*17-based phenotype – to indicate that phenotype was determined based on combined CYP2C19*2 genotype and CYP2C19*17 genotpye. Nomenclature CYP2C19*2/*17 denotes intermediate metabolizers. So, it should be clear that the wording “CYP2C19*2*17-based phenotype” is phenotype determined based on the two genotypes. This is consistent throughout the manuscript.

Comment 3. Line 36: In my opinion, it would be better to replace it with “The cytochrome P-450 isozymes 2C19 (CYP2C19)~”.

Reply. Done.

Comment 4. line 37: Replace 35 star to 39 star.

Reply. Corrected.

Reviewer 3 Report

Comments and Suggestions for Authors

In this study, authors analyzed the influence of the haplotype CYP2C:TG and its association with ischemic events and bleedings in 283 clopidogrel-treated patients from Croatia. Their preliminary results showed a tendency of higher incidence of the mentioned ADR in PM-IM vs the other metabolizer phenotypes. However, they concluded no major role of this haplotype for informing the risk of treatment failure or bleeding events in patients treated with clopidogrel.

This paper is well written. I appreciate that the authors recognize several of the limitations of the study.

I have some minor comments:

-These findings were observed in Caucasians of European (Slavic) descent. Author should mention that these results may not generalized to other populations.

-The range of age of patients is too wide 60±12 (22-80) (22-85, this is different in lane 204, please verify it). They included young and senior patients with different hepatic and renal functions. Perhaps an analysis by age groups can be performed.

-Line 126. Correct  „exposures“

-Please, define abbreviations when first time are used e.g., CVI is included in Table 1, but defined until Table 2.

-The following phrase in the title of the Table 1, I will put it down in the corresponding table foot. “Values are n (percent) or mean±SD (minimum-maximum) for age.”

-The same applies for the other titles of tables. Titles must be short and explicit. Explanation of data and abbreviations should be written in the Table foot.

-In lines 194-195 “[…]Standardized mean differences (d)<0.1 between pairs of exposure (phenotype, haplotype) levels were considered to indicate adequate balance[…]” Please include a reference for this. Authors used maximum standardize mean differences (Max d). Does Standardized Mean Difference refer measure Effect Size? Please, explain in methods.

-In the foot of Table 4, appears superscript #1 “1 Effective sample sizes…”, but this superscript is not indicated in the table.

Author Response

Comment 1. These findings were observed in Caucasians of European (Slavic) descent. Author should mention that these results may not generalized to other populations.

Reply. We agree. The fact is that generalizability of the present finding is inherently limited considering its exploratory nature (a limited sample size, possible residual confounding) – and we explicitly state this fact. However, it is true that the situation might be different in other ethnic/racial populations. We included a “Conclusions” sectin in which we address this point.

Comment 2. The range of age of patients is too wide 60±12 (22-80) (22-85, this is different in lane 204, please verify it). They included young and senior patients with different hepatic and renal functions. Perhaps an analysis by age groups can be performed.

Reply. Thank you for spotting the discrepancy between Table 1 and the textual part of the Results. The fact is that the actual age range was 22-85 years – the Table 1 entry was a typo – and has been corrected.

It is true that the age range was rather wide, but this is nothing unusual in this setting. We will use just one recent example: a recently published (Circulation 2023; 147:108-117) randomized trial (HOST-EXAM extended study) compared clopidogrel to aspirin for long-term mono-treatment after 12-18 of dual antiplatelet treatment after PCI. One of the inclusion critera was “age >20 years” and there was no upper age limit. In the present analysis, PCI patients had an “upper limit” of 80 years (at a discretion of the attending cardiologists), while no upper age limit was used for cerebrovascular patients. All patients had to be >18 years of age (the lowest age was 22 years). In the mentioned RCT, mean age in 2600 clopidogrel treated patients was 63.3±10.7 years and in aaround 2600 aspirin-treated patients, it was 63.3±10.8 years. This is closely similar to the age in the present analysis (range 22-85, 60±12). But, the main point we would like to emphasize is that the patients with different levels of “exposure” (PM/IM, NM and RM/UM, or haplotype carriers vs. non-carriers) were balanced on age by entropy balancing. I.e, they were balanced on age, sex, comorbidities etc (Table 3, Table 4) – please note that their characteristics across the subsets are virtually identical – as if they were randomized. This is exactly the purpose of entropy balancing: to achieve a balance comparable to simple randomization – but only on the measured covariates. Hence, the fact that the age range in this cohort was wide – has nothing to do with the estimate of the haplotype effect (since the impact of age was controlled). It is exactly the case as in an RCT. We agree that it would be interesting to explore whether age might be a moderator of the effect of this haplotype. However, for the present report – this is impossible. In a randomized trial, to conduct a meaningful analysis of this type, subjects would need to be randomized by strata of age (e.g., age-stratified randomization across 3 or 4 “age-bands”). In an observations study, this could be achived by performing separate balancing within each age stratum. But the present sample is too small – this was an exploratory analysis.

Comment 3. Line 126. Correct  „exposures“

Reply. Corrected.

Comment 4. Please, define abbreviations when first time are used e.g., CVI is included in Table 1, but defined until Table 2.

Reply. Corrected (added to the footnote).

Comment 5. The following phrase in the title of the Table 1, I will put it down in the corresponding table foot. “Values are n (percent) or mean±SD (minimum-maximum) for age.”

Comment 6. The same applies for the other titles of tables. Titles must be short and explicit. Explanation of data and abbreviations should be written in the Table foot.

Reply. These two comments are related. We are aware that the table titles should be explicit. The Tables should also be self-explanatory. All table titles in this manuscript were conceived in the sam way: first, there is an explicit title that also informs about the nature of the content. This is followed by a short explanation of the numerical nature of the shown data. Additional info is added if needed. Abbreviations and other explanations - if needed – are in the footnote. The journal has no explicit recommendations about what exactly should be in the title and what in the footnotes. The other two reviewers and editor(s) had no comments on table titles. Hence, we prefer to keep them as they are.

Comment 6. In lines 194-195 “[…]Standardized mean differences (d)<0.1 between pairs of exposure (phenotype, haplotype) levels were considered to indicate adequate balance[…]” Please include a reference for this. Authors used maximum standardize mean differences (Max d). Does Standardized Mean Difference refer measure Effect Size? Please, explain in methods.

Reply. Please note – the literature on methods used to achieve covariate balance between exposure levels in observational studies is extremely huge (propensity score-based methods, number of methods not requiring propensity score calculation). Similarly, literature on methods to assess whether the achieved balance is adequate is extensive. There are several methods, but the standard one and most widely recommended method is by determining standardized mean differences between exposure levels. The point about d of <0.1 indicating adequate balance is a generally and universally accepted criterion. There is no single reference to cite (one would probably need to cite several tens of references). Generally, one can use any level of d as an indicator of “adequate balance”. Not uncommonly, investigators use the cut-off of 0.2. In some rare instances, some authors have used 0.05. But 0.1 is a common and widely accepted limit, a general knowledge without any direct citation. Readers might agree that this is an adequate limit and, in this light, value the presented result, or may consider it too liberal, and disreagard the reported result – this is their choice. Some might think that this is a too conservative of a limit, other might think that is just appropriate. We, following a general and dominant practice, have chosen to use this cut-off. There is no single reference to cite – a reader might agree or not with our choice, and thus interpret the present findings. Further, “d” is calculated for pairwise differences (e.g., exposure has two levels). However, in the present results we report also on exposures with 3 levels. “Max d” is nothing else but the largest of the three “d” values obtained for three  pairwise difference (exposure level 3 vs. 1, 3 vs. 2 and 2 vs. 1). This is general knowledge that does not require further elaborations (we used the same approach in eg. Bichem Med 2024; 24:020703; Fundam Clin Pharmacol 2024; 38:351-368; Croat Med J 2023; 64:344.-353; Eur J Clin Pharmacol 2023; 79:1117-1129; Eur J Clin Pharmacol 2023; 79: 643-655; Adv Ther 2023; 40:601-618; Br J Clin Pharmacol 2023; 89:787-831; Br J Clin Pharmacol 2022; 88:2190-2202; Eur J Clin Pharmacol 2022; 78:227-236; Clin Transplant 2022; 36:e14468).

Comment 7. In the foot of Table 4, appears superscript #1 “1 Effective sample sizes…”, but this superscript is not indicated in the table.

Reply. Thank you for spotting this. Superscript was also missing in Table 3 – it was added to both tables.

Round 2

Reviewer 2 Report

Comments and Suggestions for Authors

Overall, the contents of the paper have improved according to the reviewer's comments